# Imaging Hydrogen Sulfide in Hypoxic Tissue with [^99m^Tc]Tc-Gluconate

**DOI:** 10.3390/molecules26010096

**Published:** 2020-12-28

**Authors:** Yongkyoung Kweon, Ji-Yong Park, Young-Joo Kim, Yun-Sang Lee, Jae-Min Jeong

**Affiliations:** 1Department of Nuclear Medicine, Seoul National University College of Medicine, Seoul 03080, Korea; kweon605@gmail.com (Y.K.); pjypoo@hanmail.net (J.-Y.P.); yjukim@snu.ac.kr (Y.-J.K.); wonza43@snu.ac.kr (Y.-S.L.); 2Department of Biomedical Sciences, Seoul National University College of Medicine, Seoul 03080, Korea; 3Department of Nuclear Medicine, Cancer Research Institute, Seoul National University, Seoul 03080, Korea

**Keywords:** H_2_S, technetium-99m, hypoxia, gluconate

## Abstract

Hydrogen sulfide (H_2_S) is the third gasotransmitter and is generated endogenously in hypoxic or inflammatory tissues and various cancers. We have recently demonstrated that endogenous H_2_S can be imaged with [^99m^Tc]Tc-gluconate. In the present study, we detected H_2_S generated in hypoxic tissue, both in vitro and in vivo, using [^99m^Tc]Tc-gluconate. In vitro uptake of [^99m^Tc]Tc-gluconate was measured under hypoxic and normoxic conditions, using the colon carcinoma cell line CT26, and was higher in hypoxic cells than that in normoxic cells. An acute hindlimb ischemia-reperfusion model was established in BALB/c mice by exposing the animals to 3 h of ischemia and 3 h of reperfusion prior to in vivo imaging. [^99m^Tc]Tc-gluconate (12.5 MBq) was intravenously injected through the tail vein, and uptake in the lower limb was analyzed by single-photon emission computed tomography/computed tomography (SPECT/CT). SPECT/CT images showed five times higher uptake in the ischemic limb than that in the normal limb. The standard uptake value (SUVmean) of the ischemic limb was 0.39 ± 0.03, while that of the normal limb was 0.07 ± 0.01. [^99m^Tc]Tc-gluconate is a novel imaging agent that can be used both in vitro and in vivo for the detection of endogenous H_2_S generated in hypoxic tissue.

## 1. Introduction

Gasotransmitters are small signaling molecules that can freely permeate through membranes [1]. Gaseous signaling molecules are either synthesized endogenously in an organ, tissue, or cell, or are received from outside, and the production can be induced by certain physiological or biochemical changes [2]. Gasotransmitters are a subfamily of endogenous gas molecules, including nitrogen oxide (NO), carbon monoxide (CO), and hydrogen sulfide (H_2_S), which is the most recently identified gasotransmitter [3]. H_2_S plays an important role in various physiological processes, such as regulation of inflammation, vasodilation, oxygen sensing, angiogenesis, hypoxia, and reperfusion injury [4].

H_2_S can easily be oxidized to sulfur-containing substances, such as hydrogen sulfide (HS^−^) and sulfide (S^2−^) in aqueous solution [5]. Dynamic equilibrium in the sequential reactions is achieved as follows:H_2_S ⇌ HS^−^ +H^+^ ⇌ S^2−^ + 2H^+^

Approximately 20% of the H_2_S content exists in an undissociated form, and the rest is dissociated as HS^−^ and H^+^ under physiological conditions of pH 7.4 and a temperature of 37 °C. HS^−^ can dissociate further into H^+^ and sulfide ions (S^2−^) at alkaline pH. Therefore, S^2−^ is not found in significant quantities in vivo. However, there is no exact ratio for the level of individual components of all three molecules of sulfide. H_2_S can be produced endogenously in mammalian tissues through both non-enzymatic and enzymatic pathways [2,5,6]. The nonenzymatic pathway occurs when cysteine thiols interact with H_2_S to form stable persulfides at pH 8.4 [6]. To date, the contribution of the nonenzymatic pathway to the overall H_2_S production inside the cells remains unclear. For the enzymatic pathway, there are three main enzymes, namely, cystathionine β-synthase (CBS), cystathionine γ-lyase (CSE), and 3-mercaptopyruvate sulfurtransferase (3-MST) [7].

Recent research has demonstrated that the CSE protein is localized in the cytosol under normal conditions, but translocates into the mitochondria under hypoxic conditions, where it metabolizes L-cysteine to produce H_2_S [7,8]. Production of H_2_S is regulated by changes in the partial pressure of oxygen, which increases under hypoxic conditions [8]. H_2_S also interacts with different membrane ion channels to regulate the vascular smooth muscle [9]. Increased H_2_S levels may cause pre-constricted blood vessels to expand, thereby increasing the supply of oxygen under oxygen-deprived conditions [10].

Common methods of detecting H_2_S include the use of methylene blue assay, ion-selective electrodes, amperometric sensors, or gas chromatography [11]. Although no adequate quantification method has been developed to detect H_2_S inside intact living organisms [12], there are several reports of in vivo imaging of H_2_S using fluorescence and luminescence. However, fluorescence probes lack sensitivity and depth penetration, due to background autofluorescence and light scattering [13]. Therefore, in the present study we used radionuclides for imaging of endogenously produced H_2_S.

Single-photon emission computed tomography (SPECT) is an in vivo imaging modality. When a radiotracer emitting gamma-ray is injected into the body, it can be measured directly by SPECT cameras [14]. Although there are many radioisotopes available for in vivo imaging, technetium-99m is the most commonly used isotope for diagnostic purposes [15]. ^99m^Tc has a short half-life of 6 h and is inexpensive and available in many hospitals. It can be obtained conveniently from ^99^Mo/^99m^Tc generators. The use of a ^99m^Tc-labeled agent for diagnosis of hypoxia is therefore practical in the clinical setting.

The first reported radionuclide for imaging H_2_S was a ^64^Cu-labeled cyclen, which was imaged using positron emission tomography (PET) [16]. However, ^64^Cu is only available in limited institutions that have expensive cyclotrons, and the production cost of ^64^Cu is high. Recently, we developed ^99m^Tc-labeled agents for the quantification of H_2_S in vivo [17]. Various α-hydroxy acids such as D-gluconic acid, D-glucaric acid, and glucoheptonic acid, can be labeled with ^99m^Tc. These ^99m^Tc-labeled agents form an insoluble fraction in the presence of H_2_S, which enables imaging of endogenously produced H_2_S.

In the present study, we evaluated [^99m^Tc]Tc-gluconate for imaging of H_2_S under hypoxic conditions, since it showed the highest percentage of insoluble fraction formation with H_2_S. We also compared the results of in vivo imaging of endogenously produced H_2_S using a fluorescent probe with imaging using [^99m^Tc]Tc-gluconate.

## 2. Results

### 2.1. ^99m^Tc-Labeling of Gluconate and Diethylenetriaminepentaacetic Acid (DTPA)

Stannous chloride was used as a reducing agent for the labeling of gluconate and DTPA with ^99m^Tc. Both reagents demonstrated near quantitative labeling yield (99.8% ± 0.1%, 399.2 MBq).

### 2.2. Formation of Insoluble Fraction with the Presence of H_2_S

NaHS solution was used instead of H_2_S gas for convenience, as it generates H_2_S in an aqueous solution. [^99m^Tc]Tc-Gluconate and [^99m^Tc]Tc-DTPA were incubated with 0.1 mM NaHS in 0.1 M phosphate buffer (pH 7.4) for 10 min at 37 °C. The reactants were analyzed using instant thin layer chromatography (ITLC)/saline. [^99m^Tc]Tc-Gluconate formed an insoluble fraction (88.9% ± 2.4%) in the presence of H_2_S; however, [^99m^Tc]Tc-DTPA did not form an insoluble fraction (Figure 1).

### 2.3. In Vitro Hypoxic Cancer Cell Uptake of [^99m^Tc]Tc-Gluconate

Cell uptake measurements of [^99m^Tc]Tc-gluconate and [^99m^Tc]Tc-DTPA as the control were performed in a hypoxia- and normoxia-conditioned colon carcinoma cell line, CT26, for up to 2 h. The radioactivity of the sample, indicated as counts per minute (CPM), was divided by the total protein amount and expressed as mean ± SD for triplicate measurements. At 120 min of incubation, [^99m^Tc]Tc-gluconate uptake in hypoxic and normoxic CT26 cells increased 109,772.48 ± 6889.49 and 25,588.32 ± 1886.07 CPM/mg, respectively (Figure 2). In contrast, [^99m^Tc]Tc-DTPA uptake only reached 5198.67 ± 466.80 and 4770.28 ± 167.79 CPM/mg, respectively, at 120 min (Figure 2). [^99m^Tc]Tc-Gluconate demonstrated significantly increased uptake in hypoxia-conditioned cancer cells, and slightly increased uptake in normoxic cancer cells. However, the control did not show any increased uptake both in hypoxic and normoxic cancer cells.

### 2.4. In Vivo Detection of Endogenous H_2_S Using a Fluorescence Probe in a Mouse Model

To compare the H_2_S production in normal and ischemic hind limbs, an H_2_S-detecting fluorescence probe (HSip-1) was injected intravenously at different time points during reperfusion. The highest fluorescence intensity in the ischemic limbs was detected after 3 h of reperfusion, which indicated that H_2_S production was highest at 3 h after reperfusion. The H_2_S detection probe (HSip-1) mainly accumulated in the ischemic leg, while a relatively less intense fluorescence signal was observed in the normal leg (Figure 3). The signal decreased as the reperfusion time increased beyond 3 h. At 24 h of reperfusion, fluorescence intensities were almost the same in both the normal and ischemic legs (Figure 3).

Immunofluorescence staining and confocal imaging were performed with hindlimb sections after 3 h of reperfusion. The fluorescence intensities of EF5 (hypoxia detection) and HSip-1 (H_2_S detection) in the ischemic leg tissue were higher than those in the normal leg tissue (Figure 4).

### 2.5. In Vivo SPECT/CT in a Tourniquet-Induced I/R Mouse Model

To confirm that [^99m^Tc]Tc-gluconate could be used as an H_2_S imaging agent in vivo, [^99m^Tc]Tc-gluconate or [^99m^Tc]Tc-DTPA was injected intravenously into a tourniquet-induced ischemia mouse model. The legs were imaged with SPECT/CT 1 h after the injection. The images showed markedly higher uptake of [^99m^Tc]Tc-gluconate in the ischemic leg than that in the normoxic leg (Figure 5A). The standard uptake value (SUVmean) of the ischemic limb was 0.39 ± 0.03, while that of the normal limb was 0.07 ± 0.01. [^99m^Tc]Tc-DTPA showed slightly increased uptake in the ischemic leg than that in the normal leg (Figure 5B). The data suggest the use of [^99m^Tc]Tc-Gluconate as a H_2_S detecting agent in vivo.

## 3. Discussion

Hydrogen sulfide (H_2_S) has been known as a toxic molecule in biological systems for centuries. Recent studies have elucidated potential regulatory functions of H_2_S, which are similar to those of other gasotransmitters such as nitric oxide and carbon monoxide. When hypoxic conditions are induced, H_2_S production increases via changes in the partial pressure of oxygen [18]. Since hypoxia is a common characteristic of solid tumors, it is necessary to detect and image endogenously produced H_2_S under hypoxic conditions.

After injecting an H_2_S detection probe (HSip-1), higher fluorescence intensity was observed in the ischemic-reperfusion injury legs compared to that in normal legs at every time point except at 24 h of reperfusion, thereby demonstrating that H_2_S formation increased after 3 h of reperfusion. A previous study measured muscle blood flow by placing a transonic flow probe in the gastrocnemius muscles connected to a laser Doppler blood flowmeter. Blood flow to the gastrocnemius muscle was measured during the ischemia-reperfusion for 3 h of ischemia and 4 h of reperfusion. When a tourniquet was applied to the limb, blood flow decreased to 2% of the baseline level and remained at the same level for 3 h of ischemia. When the limb was released, the blood flow rapidly increased to approximately 50% of the baseline level and declined to a steady state of 30% of the baseline level [19]. The results demonstrated that the fluorescence probe could not reach the gastrocnemius muscle after 1 h of reperfusion, as the blood vessels were clogged due to application of the tourniquet. Further measurements of blood flow during SPECT/CT imaging is warranted to elucidate the effects of reperfusion on probe movement through the limbs. Although no exact mechanisms for the formation of H_2_S during reperfusion have been proposed thus far, in the present study, blood flow was consistent during the reperfusion time, and formation of H_2_S increased at 3 h of reperfusion. In the present study, the fluorescence intensity in vivo imaging showed that H_2_S was produced under hypoxic conditions and could be detected using fluorescence probes in vivo. We also used radionuclides for detecting H_2_S. In the present study, a technetium-99m (^99m^Tc)-labeled agent was used because it is frequently used in radiology departments, is easy to access, and can be generated inexpensively. There are several advantages of using ^99m^Tc, such as its rapid diagnosis and 6 h half-life, which implies that the radioisotopes decay quickly, and cause relatively less damage and no side effects [20].

Gluconic acid was successfully labeled with ^99m^Tc (Figure 4), and [^99m^Tc]Tc-gluconate formed with NaHS. In a previous study by our group, [^99m^Tc]Tc-gluconate formed the highest percentage of insoluble fraction, which was not identical to Tc_2_S_7_ [17]. However, identification of the chemical structure of the insoluble fraction remains challenging. One possible reason for the formation of the insoluble fraction is the formation of insoluble polydisulfide [Tc_3_(μ^3^ – S)(S_2_)_3_(S_2_)_3/3_]*_n_* by the trimerization of insoluble sulfide Tc(IV)S_2_ [21]_._ The chemical structure of [^99m^Tc]Tc-gluconate is two gluconate molecules forming a complex with Tc(V)O. The complex contains a Tc = O core and two gluconate ligands (oxobis(gluconto)technetate(V) anion (net charge: −1) in aqueous solution) [22]. In our previous study, we determined the chemical structure of [^99m^Tc]Tc-glucoheptonate, and because of the molecular similarity of glucoheptonate and gluconate, [^99m^Tc]Tc-gluconate should demonstrate a similar structure. If this structure is correct for [^99m^Tc]Tc-gluconate, in the presence of H_2_S, Tc(V) can be reduced to Tc(IV) and precipitate, which will ensure feasibility of imaging. Another possibility is that ^99m^Tc can be fixed by transchelation with the sulfhydryl groups of proteins [23,24]. Further studies are warranted to identify the chemical structure of the insoluble fraction, and to determine the mechanisms of its formation.

Multiple studies have shown that [^99m^Tc]Tc-gluconate can be used for cancer, inflammation, and cerebral and myocardial ischemia imaging; however, the mechanism for its uptake has not been elucidated thus far. In the present study, [^99m^Tc]Tc-gluconate uptake increased when hydrogen sulfide was produced under hypoxic conditions. This is possibly due to the formation of an insoluble fraction, thereby enabling imaging of H_2_S.

An additional benefit of using [^99m^Tc]Tc-gluconate as a H_2_S imaging agent compared to fluorescence imaging probes is that [^99m^Tc]Tc-gluconate is already approved as a radiopharmaceutical, and has already been used in some countries. The application of [^99m^Tc]Tc-gluconate in a clinical setting for H_2_S detection should therefore be approved faster than that for fluorescence probes. Ultimately, a test for H_2_S is warranted in the clinical setting and our findings support the use of [^99m^Tc]Tc-gluconate, both in vivo and in vitro.

## 4. Materials and Methods

### 4.1. Solvents and Chemicals

BCA Protein Assay Kit was purchased from Pierce Co (Rockford, IL, U.S.A.). All chemicals, reagents and solvents used were of analytical grade, and were purchased from Sigma Aldrich and TCI (Tokyo, Japan). ^99m^Tc was obtained from a ^99^Mo/^99m^Tc-generator purchased from Sam Young Unitech Co. (Seoul, Korea). Instant thin-layer chromatography (ITLC) plates were purchased from Aglient Technologies (Santa Clara, CA, USA). The Bio-Scan AR-2000 scanner (Bioscan Co., Wilmington, MA, USA) was used for ITLC plate scanning. SPECT/CT animal images were acquired using NanoSPECT/CT^Plus^ (Mediso, Budapest, Hungary) and analyzed with the InVivoScope program.

### 4.2. ^99m^Tc-Labeling of Gluconate and DTPA

Distilled water purged with N_2_ gas was used for the labeling experiment. For the [^99m^Tc]Tc-gluconate labeling, 100 µL of 0.3 M D-gluconic acid sodium salt was prepared and added to 10 µL of sodium ascorbic acid (25 mg/mL) and 50 µL of SnCl_2_ 2H_2_O (1 mg/mL in 0.05 M HCl), and mixed by vortexing. Approximately 370 MBq (140 µL) of [^99m^Tc]NaTcO_4_ eluted in normal saline obtained from the generator was added to the mixture at room temperature for 20 min (pH 4.5).

Radiochemical quality control testing of [^99m^Tc]Tc-gluconate using two instant thin-layer chromatography strips was performed via radio TLC. A drop of the labeled mixture was placed near the bottom of each ITLC strip and then placed in acetone or saline. As the solvent migrates upward, soluble radiochemical species are carried upward along with the solvent, while insoluble compounds remain at the origin. The radioactivity at each location on the strip was measured using a radio TLC scanner (Bioscan Co., WY, USA). When the mixture was eluted with acetone, reduced-hydrolyzed ^99m^Tc impurity and ^99m^Tc-labeled gluconate remained at the origin and unlabeled free ^99m^Tc moved upward to the solvent front. In contrast, when the mixture was eluted with saline, only reduced-hydrolyzed ^99m^Tc remained at the origin and [^99m^Tc]Tc-gluconate moved to the solvent front.

For the [^99m^Tc]Tc-DTPA labeling, a DTPA kit vial (Mallinckrodt, The Netherlands) was obtained from Seoul National University Hospital. Four milliliters of freshly eluted [^99m^Tc]NaTcO_4_ (400 MBq) obtained from the generator was added to the DTPA kit. Further, the labeled product was analyzed by ITLC and eluted with acetone or saline solution. Similarly, when it was eluted with acetone, labeled [^99m^Tc]Tc-DTPA remained at the origin, while labeled [^99m^Tc]Tc-DTPA moved to the solvent front when it was eluted with saline.

### 4.3. Formation of Insoluble Fraction in the Presence of H_2_S

One hundred-microliter samples of [^99m^Tc]Tc-gluconate and [^99m^Tc]Tc-DTPA were mixed with one hundred microliter of 0.2 mM NaHS in 0.2 M sodium phosphate buffer (pH 7.4). The mixtures were vortexed and incubated at 37 °C for 10 min, spotted onto ITLC plates, and eluted with saline. The ITLC plates were scanned using the Bio-Scan AR-2000. The insoluble fraction was analyzed by calculating the percentage of remaining radioactivity at the origin.

### 4.4. Cell Culture

CT26 (colon carcinoma cell line) cells were grown in the DMEM medium (Gibco, Grand Island, NY, USA) containing 10% (*v*/*v*) fetal bovine serum (Gibco, Grand Island, NY, USA) and 1% penicillin/streptomycin (Invitrogen, Grand Island, NY, USA). Cells were incubated at 37 °C in a humidified incubator under a 5% CO_2_ atmosphere.

### 4.5. In Vitro Hypoxic Cell Culture Uptake of ^99m^Tc-Gluconate

CT26 cells were purchased from the Korean Cell Line Bank, cultured in the DMEM medium (High glucose, Gibco, Grand Island, NY, USA), and incubated at 37 °C and 5% CO_2_ in a humidified incubator.

Cells were cultured under hypoxic conditions (95% N_2_ and 5% CO_2_) and normoxic conditions (95% air and 5% CO_2_) for measurement of [^99m^Tc]Tc-gluconate and [^99m^Tc]Tc-DTPA uptake. A suspension of 5 × 10^6^ cells/mL was produced in N_2_-purged media in sterile glass vials. Cells were incubated for 4 h under hypoxic and normoxic conditions. [^99m^Tc]Tc-gluconate was diluted in a serum-free, N_2_-purged DMEM medium and 1 mL was added into vials. The vials were incubated in a humidified incubator at 37 °C and 5% CO_2_. At 30 min, 1 h, and 2 h, 200 μL of each suspension was collected into a 5 mL test tube. The cells were washed and centrifuged three times with cold Hank’s balanced salt solution (HBSS), lysed in 0.2 mL of 1% SDS, and transferred into new 5 mL test tubes. The radioactivity of the cells was determined using a γ-counter. A BCA protein assay was performed the next day to determine total cell protein, according to the manufacturer’s instructions.

### 4.6. Ischemia-Reperfusion Mouse Modeling for In Vivo Imaging

Tourniquet-induced ischemia-reperfusion was performed according to a previously reported method [25,26]. Mice were anesthetized with an anesthetic cocktail consisting of 0.1 mg/g alfaxalone and 0.01 mg/g xylazine, administered via intraperitoneal injection (0.005 mL/g body weight).

Anesthesia was maintained throughout the experiments with an additional anesthetic cocktail (0.05 mL) as per requirements. Under anesthesia, mouse fur was completely removed from both hind limbs using an electric shaver and hair removal cream. The animals were restrained on a heating pad to maintain their body temperature at 37 °C.

Unilateral hind limb ischemia was induced by placing an orthodontic rubber band at the hip joint. After 3 h of ischemia, the orthodontic rubber band was released, and reperfusion was performed.

### 4.7. In Vivo Detection of Endogenous H_2_S by Fluorescence and SPECT/CT Imaging in a Mouse Model of Ischemia

To detect endogenously produced H_2_S in the ischemic hindlimb, 100 μL of HSip-1 (200 μM) was intravenously injected after 0, 1, 3, 12, and 24 h of reperfusion. After 1 h of circulation, the mice were sacrificed, and the fluorescence signal intensities of the left and right limbs were measured. Fluorescence signals were measured using the Lumina II fluorometer (Perkin Elmer, Waltham, MA, USA). Fluorescence images were acquired with an exposure time of 20 s, medium binning, 2 f/stop, and an open filter. Data were analyzed using the Living Image software (version 2.5)

For SPECT/CT imaging, after 3 h of reperfusion, 12.5 MBq [^99m^Tc]Tc-gluconate or 12.7 MBq [^99m^Tc]Tc-DTPA in 100 μL saline was intravenously injected. Only the lower limb was imaged with SPECT/CT by covering the upper limb with a 2 mm thick lead plate.

The scanning parameters for the whole-body imaging modality used a γ-ray energy window of 140 keV ± 10%, a matrix size of 256 × 256, an acquisition time of 5 s per angular step of 18°, and a reconstruction algorithm of ordered subset expectation maximization with 9 iterations. For integrated CT, a tube voltage of 45 kVp, an exposure time of 1.5 s per projection, and a reconstruction algorithm of cone-beam filtered back-projection was used. The SPECT/CT images were represented using the same scale conditions in InVivoScope on the nanoScan-SPECT/CT.

### 4.8. Immunofluorescence Staining of Hindlimb Sections

A two-hundred-microliter volume of 10 mM EF5 (hypoxia marker, #CS222743, Millipore) was injected with HSip-1 (H_2_S marker, SB21, Dojindo Molecular Tech.). Both the ischemic and normal hind limbs were extracted immediately after analysis on the Lumina II fluorometer (Perkin Elmer, Waltham, MA, USA). Limbs were embedded in optimal cutting temperature (OCT) compound (Leicabiosystems, Richmond, IL, USA), snap-frozen, sectioned at 8 μm thickness using a cryostat, and placed on slides. Sections were dried for 30 min at room temperature, fixed with acetone at −20 °C for 10 min, and then allowed to dry for 30 min at room temperature. Fixed slides were washed three times with PBS for 5 min. After washing, all sections were marked with a PAP pen (hydrophobic pen), and blocking solution was added. The slides were placed in a staining tray containing a small amount of water covered with a lid overnight at 4 °C. After the blocking solution was removed, all slides were washed and rinsed with 1× ttPBS (1× PBS with 0.3% Tween 20, and 2 mM sodium azide).

Staining solution (Cy3-conjugated anti-EF5 (75 μg/mL), 100 µL) was added to slides for 6 h in a tin-foil-covered staining tray with a lid at 4 °C. After staining, all slides were washed three times with cold PBS for 40 min in the dark. Mounting was performed with the ProlongTM Gold antifade reagent with DAPI (Invitrogen, USA). Fluorescence images were acquired the next day using a confocal laser scanning microscope (Leica TCS SP8, Wetzlar, Hesse, Germany).

### 4.9. Confocal Imaging

Confocal imaging was used to analyze the fluorescence signal in the samples. After performing immunofluorescence staining, samples were washed with PBS three times, and the samples were mounted with the ProLong^TM^ Gold antifade reagent with DAPI (Invitrogen, Grand Island, NY, U.S.A) and covered with a cover slide. The confocal imaging samples were stored at 4 °C and imaged the next day. Fluorescence signals were detected using a confocal laser scanning microscope (Leica TCS SP8m Wetzlar, Hesse, Germany) in the specific range of wavelength (DAPI; 401–480, FITC; 495–519 Cy3; 550–570). Fluorescence intensities were analyzed using the LAS X system (Leica Microsystems, Wetzlar, Germany).

### 4.10. Statistical Analysis

Quantitative data are expressed as mean ± SD. Mean values were compared using the Student’s *t*-test in Excel (Microsoft, Redmond, WA, U.S.A.) or GraphPad Prism (GraphPad Software, Inc., San Diego, CA, USA).

## 5. Conclusions

Endogenously produced H_2_S in hypoxic tissue can be imaged using [^99m^Tc]Tc-gluconate both in vivo and in vitro.

## Figures and Tables

**Figure 1 molecules-26-00096-f001:**
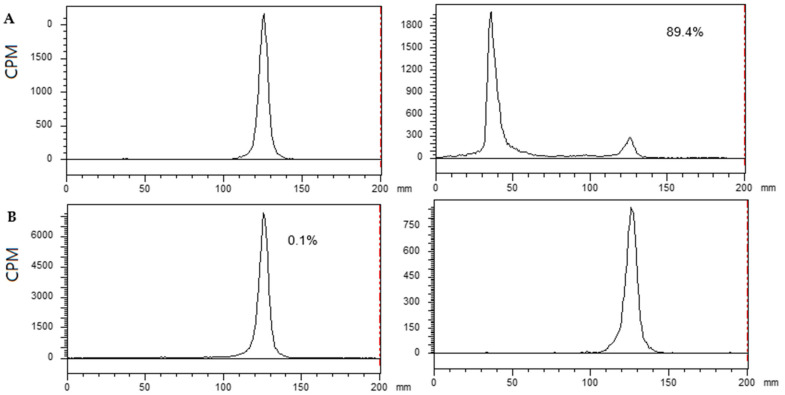
Comparison of insoluble fraction formation. Instant thin layer chromatography-silica gel (ITLC-SG) data eluted with saline. ^99m^Tc-labeled agents were subjected to reaction with NaHS at 37°C for 10 min at pH 7.4. The radioactivity remaining at the origin was calculated as the insoluble fraction. (**A**) [^99m^Tc]Tc-Gluconate (left: before reaction; right: after reaction) and (**B**) [^99m^Tc]Tc-DTPA (left: before reaction; right: after reaction).

**Figure 2 molecules-26-00096-f002:**
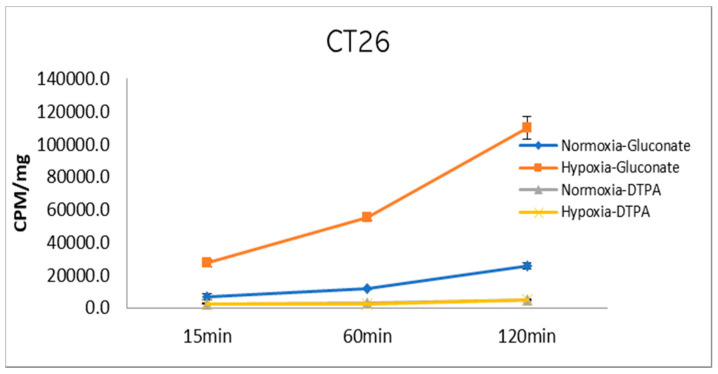
In vitro uptake of [^99m^Tc]Tc-gluconate and [^99m^Tc]Tc-DTPA. Cell uptake was measured under hypoxic and normoxic conditions. Cells were harvested at 15, 60, and 120 min, and the radioactivity was measured using a gamma counter. The value was expressed as CPM/mg of the total cellular protein. (*n* = 3).

**Figure 3 molecules-26-00096-f003:**
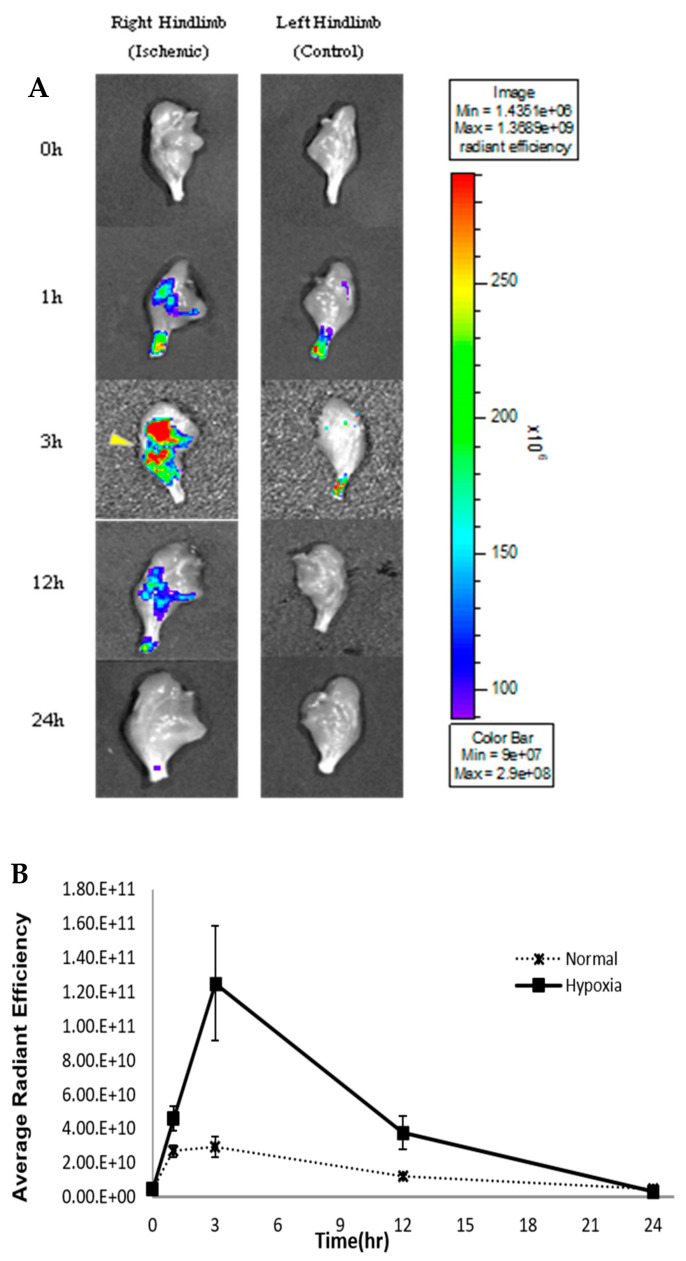
In vivo fluorescence imaging of endogenous H_2_S by fluorescence probe in a rat model. (**A**) Fluorescence images (Lumina II) at the following time points: 0, 1, 3, 12 and 24 h. Left legs show normal and right legs show hypoxic conditions (*n* = 4). Yellow triangle represents hypoxic region. (**B**) Fluorescence intensity of both normal and ischemic regions across 24 h.

**Figure 4 molecules-26-00096-f004:**
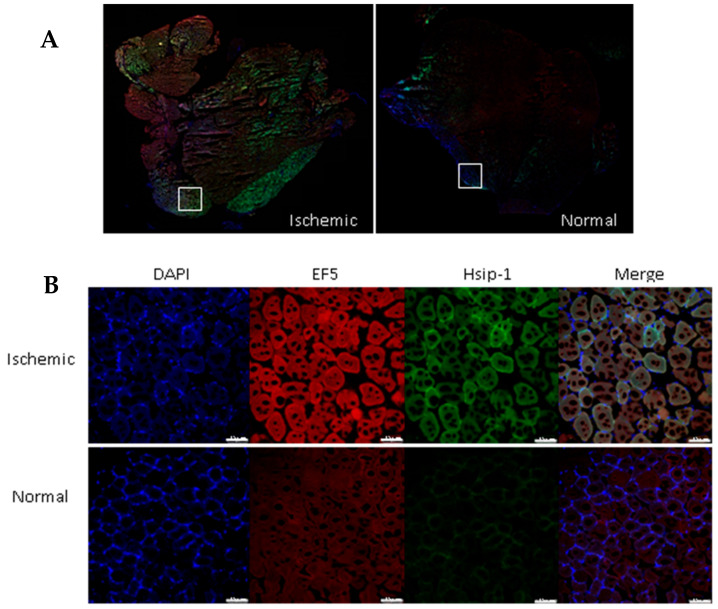
Immunofluorescence staining of hindlimb sections. (**A**) Fluorescence images of ischemic and normal tissue. Tissue imaging was performed with a confocal microscope. Original magnification: 100×. (**B**) Red indicates EF5 (hypoxia detection), green indicates HSip-1 (H_2_S detection), and blue indicates DAPI (nuclei). Original magnification: 400×; scale bars, 75 µm.

**Figure 5 molecules-26-00096-f005:**
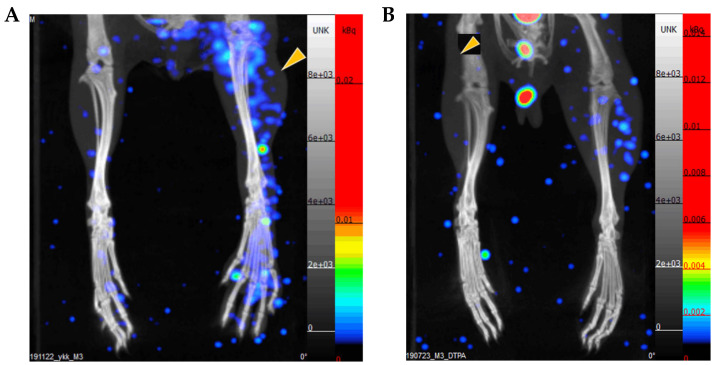
[^99m^Tc]Tc-Gluconate and [^99m^Tc]Tc-DTPA imaging of an ischemia reperfusion mouse model. BALB/c mice received 3 h of tourniquet-induced ischemia modeling after 3 h of reperfusion (yellow triangle). Thereafter, 12 MBq of (**A**) [^99m^Tc]Tc-gluconate or (**B**) [^99m^Tc]Tc-DTPA was intravenously injected via tail vein. After 1 h, legs were imaged by SPECT/CT. (**C**) Average standard uptake value (SUV) representing uptake in both normal and ischemic limbs. Statistical significance was determined using an unpaired Student’s *t*-test. ** *p* < 0.05, *** *p* < 0.01.

## Data Availability

Data sharing not applicable.

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
