# Peer review of "Imaging Hydrogen Sulfide in Hypoxic Tissue with [99mTc]Tc-Gluconate"

_molecules, 2020, doi:10.3390/molecules26010096_

Round 1
Reviewer 1 Report
The manuscript describes the radiosynthesis and preclinical evaluation of the gastrotransmitter H2S PET tracer [99mTc]Tc-gluconate. This gastrotransmitter is a target of great interest, and only a 64Cu-radiolabelled tracer has been described to image it using nuclear medicine imaging, which gives to this manuscript a particular importance, considering the current reach of scintigraphy and SPECT imaging with technetium-99m worldwide.
The manuscript is well written and structured and the results encourage the further evaluation of this radiotracer. Perhaps, the authors should considere to image hypoxic tumours in vivo (not necessary for the acceptance of the present manuscript) and compare it with current established radiotracers for imaging hypoxic tumours.
Some minor corrections should be corrected before publication of this manuscript:
- Page 8, line 277: 99Mo instead of 99M
- Page 8, line 285: I assume you mean "Distilled water purged with N2"
- Page 7, lines 257-258: I think some word is missing here to give sense to the sentence. Maybe under the presence.
In addition, it would be good to show also absolute radiochemical yields (i.e. in MBq) in the results section.
Author Response
The manuscript describes the radiosynthesis and preclinical evaluation of the gastrotransmitter H2S PET tracer [99mTc]Tc-gluconate. This gastrotransmitter is a target of great interest, and only a 64Cu-radiolabelled tracer has been described to image it using nuclear medicine imaging, which gives to this manuscript a particular importance, considering the current reach of scintigraphy and SPECT imaging with technetium-99m worldwide.
The manuscript is well written and structured and the results encourage the further evaluation of this radiotracer. Perhaps, the authors should considere to image hypoxic tumours in vivo (not necessary for the acceptance of the present manuscript) and compare it with current established radiotracers for imaging hypoxic tumours.
ANSWER. We greatly appreciate the reviewer for kind comments, and we will further investigate imaging of hypoxic tumours in the future study.
Some minor corrections should be corrected before publication of this manuscript:
- Page 8, line 277: 99Mo instead of 99M => corrected
- Page 8, line 285: I assume you mean "Distilled water purged with N2" => corrected
- Page 7, lines 257-258: I think some word is missing here to give sense to the sentence. Maybe under the presence. => added "in" before "the presence"
In addition, it would be good to show also absolute radiochemical yields (i.e. in MBq) in the results section. => we added 399.2 MBq in the parenthesis
Reviewer 2 Report
see attached file please

Author Response
This manuscript investigates the application of [99mTc]Tc-gluconate, one of the oldest known technetium-99m-based-radiopharmaceutical, for the imaging of the gasotransmitter hydrogen sulfide, generated in hypoxic tissue. It provides an explanation for the mechanism of detection, via the formation of insoluble fractions with the technetium-99m-labeled agent, and demonstrates its successful application for the in vivo imaging in H2S depended context with the expected specific signal. It is a valuable contribution to this field and highlights [99mTc]Tc-gluconate as a new potential agent for in vivo H2S imaging.
The following minor corrections should be done.
ANSWER: We corrected all the typographic errors as indicated in the table.
The final concentrations across the different experiments should be stated to improve comparability for the reader.
ANSWER: We corrected the concentration of NaHS to its final concentration.
We approve this work, yet it is noteworthy to mention a few aspects for the authors to consider, which could strengthen the paper. If any of these points are already investigated, by the group previously, present in the literature or subject to further research they could be mentioned/referred to further increase the understanding for the reader, elude misunderstanding and highlight the relevance of the findings.
ANSWER: We greatly appreciate the reiewer’s kind comments.
In section 2.2., the authors demonstrate that [99mTc]Tc-gluconate forms a fraction with the aqueous solution of NaHS. This is sufficient as proof of principle. But as the paper attempts to establish [99mTc]Tc-gluconate as imaging agent for H2S it would benefit a lot from providing further information about sensitivity of the agent. The incubation experiment could be repeated with the compound and different NaHS concentrations showing the concentration range in which formation of insoluble fraction and detection of H2S happens (and whether both are interdependent). This could prove even more important as physiological concentrations of H2S are subject to debate: PMID: 18799635. As H2S concentrations in the range 5-30 μM are already considered toxic see: PMID: 32679888 the used concentrations of 0.2 mM NaHS generating H2S might be too high unnecessary undermining the relevance of the fraction for biological probes.
ANSWER: The final concentration of NaHS was 0.1 mM. However, H2S concentration might be much lower than that due to equilibrium state. And, although blood level might be lower than the toxic level, the concentration in special tissue, which should be imaged, can be higher than that.
The authors showed in section 2.3. increased uptake of [99mTc]Tc-gluconate by hypoxia conditioned cells. For the reader it might not be completely clear why DTPA has been chosen as control. Ideally this experiment could have been accompanied by cell viability measurements showing that neither hypoxia conditions nor application of the labelled compound produce adverse effects highlighting the safety of the compound and the relevance of the uptake in living cells. In case of H2S measurements were performed parallel to the cell culture experiments even if the data is not shown we would mention it just for the reader to know that hypoxia conditioning induced H2S in these cells and their production correlates with [99mTc]Tc-gluconate uptake.
ANSWER: Unlike 99mTc-gluconate, DTPA is very stable and doesn’t form insoluble fraction with H2S. So, it was used as a control in this experiment as well as the previous experiment ref [19]. If the cells were damaged by hypoxia, then the uptake might be decreased due to the decreased cell number. However, the uptake increased in hypoxic cells which represents that the cell damage might be negligible.
Similar experiments have been conducted by Ballinger et al. (PMID 8381475) in CHO cells. The authors there demonstrated an increased accumulation of [99mTc]Tc-gluconate under hypoxic conditions. They also observed a gradual accumulation under aerobic conditions. Based on Figure 2 in your manuscript this gradual accumulation is also present in CT26. It could be at least mentioned as significant finding.
ANSWER: We changed the sentence that “[99mTc]Tc-Gluconate demonstrated significantly increased uptake in hypoxia-conditioned cancer cells, and slightly increased uptake in normoxic cancer cells. However, the control did not show any increased uptake both in hypoxic and normoxic cancer cells.” In page 3 line 126-127.
In page 6 line 205-206 and also later in the discussion part (page 7 line 264-265) the authors conclude that [99mTc]Tc-gluconate can be used as agent for the imaging of endogenously produced H2S. This claim might be too strong based on the data provided alone and should be toned down. Instead we would suggest: the data suggests/supports the use of [ 99mTc]Tc-gluconate as a hydrogen sulfide detecting agent in vivo.
ANSWER: We changed the sentence into “The data suggest the use of [99mTc]Tc-Gluconate as a H2S detecting agent in vivo.” as recommended.
We would like to elaborate here, why we would prefer the rather precautious expression.
Ideally to make the initial claim we have to prove that [99mTc]Tc-gluconate exclusively detects H2S. While the manuscript provides a mechanism how H2S detection happens, it does not exclude the possibility of interaction with other molecules. If there are any experiments in the literature present that helps to exclude alternative targets, referring to them would benefit the message of the paper.
Alternatively proving that the source of the signal is mainly coming from in vivo H2S production and showing a dose dependent signal increase could achieve the same. As it is, it is hard to uncouple the relation of hypoxia and H2S production based on the data provided. So while likely, we still cannot tell, whether the increased uptake of the labelled compound is due to
- increased H2S induced by hypoxia or
- by hypoxia related effects independent of generated H2S or
- self induced effects of our labelled compound (as H2S is discussed in context of permeability (PMID: 28148499))
- or a mixture of the cases above.
We understand, that unrevealing of these is potential subject of future research. The presented results are promising and we wish the authors success for your future research.
The authors present here an original work with high actuality. We recommend this manuscript for publication after minor revisions.
ANSWER: Many thanks again for kind comments.
Reviewer 3 Report
This manuscript describes the evaluations of 99mTc-gluconate for endogenous H2S imaging. The authors have previously published on the radiotracers to detect H2S. The radiotracer in this study suggested in vitro and in vivo detection of H2S. The study results are very interesting and, thus, of interest to the reader of Molecules. However, there are a number of aspects which should be improved. The following comments could help to improve the organization and quality of the manuscript.
1) In vitro and in vivo specificity for H2S was not evaluated. Can the author perform an experiment demonstrating specificity, such as coinjection with an H2S inhibitor? Hypoxia or ischemia induces not only H2S but also other many substances. In the discussion section, the authors describe that mechanisms how 99mTc-gluconate formed insoluble fractions are unclear. Can 99mTc-gluconate form insoluble fractions under other conditions even in the absence of H2S? Please comment.
2) It is recommended to provide biodistribution data for whole body. Nonspecific accumulation in normal organs and tissues, if any, might raise concerns about specific imaging.
3) Figure 4B. It is difficult to see fluorescent intensity from images. It is recommended to provide quantified values of fluorescent intensity.
4) Please provide ITLC data of 99mTc-gluconate and 99mTc-DTPA without NaHS.
5) Please discuss comparison of H2S with other hypoxia markers.
6) Values described in Abstract are missing in the main manuscript. Eg, 'up to 40%' and 'SUVmean'. Please also include these values in the main manuscript.
7) Line 82. ‘99mTc’ is not expressed correctly. Also in other sections such as Line 284. Please check.
8) Symbol ‘micro’ should be expressed correctly, not ‘u’.
9) Figure 1. Please include unit of vertical axis.
10) Line 109. Figure 1, not Figure 11.
11) Why did the author use CT26 cell line? Are there any specific reasons? Please comment.
12) Cell uptake should be corrected (divided) by total radioactivity applied to cells. The values are generally expressed as ‘% initial activity/mg protein’.
13) Line 119. Figure 2, not Figure 7.
14) Lines 125-127 seem to belong to a caption of Figure 2. Please move the sentences to the correct position.
15) Figure 3A. Some words are hidden by images. Please check.
16) Figure 3A and 5A. Please include an explanation what a yellow triangle indicates.
17) Names of devices do not have to be provided in the result section. Include them only in the method section.
18) Line 205. Figure 5B, not Figure 10B.
19) Figure 5A and B. Clear which leg is normal or ischemic.
20) Lines 211-212. Did the author intravenously inject the radiotracer directly into legs? Was that not via tail vein?
21) Line 277. 99Mo/99mTc-generator, not 99M/99mTc-generator.
22) In the discussion section, some repetitions of the introduction section are found. Eg, about disadvantages of fluorescent probes and advantages of 99mTc. Please remove.
23) Line 247. Fig 4 seems to be incorrect.
24) Lines 330-331. Include a manufacture name of a BCA assay (kit).
Author Response
This manuscript describes the evaluations of 99mTc-gluconate for endogenous H2S imaging. The authors have previously published on the radiotracers to detect H2S. The radiotracer in this study suggested in vitro and in vivo detection of H2S. The study results are very interesting and, thus, of interest to the reader of Molecules. However, there are a number of aspects which should be improved. The following comments could help to improve the organization and quality of the manuscript.
1) In vitro and in vivo specificity for H2S was not evaluated. Can the author perform an experiment demonstrating specificity, such as coinjection with an H2S inhibitor? Hypoxia or ischemia induces not only H2S but also other many substances. In the discussion section, the authors describe that mechanisms how 99mTc-gluconate formed insoluble fractions are unclear. Can 99mTc-gluconate form insoluble fractions under other conditions even in the absence of H2S? Please comment.
ANSWER: We are sorry that we missed inhibitor study, which was performed in our previous report using inflammation imaging (ref [19], Figure 5). However, it might be difficult to block all the H2S synthetic pathway completely in vivo. It has been proved that [99mTc]Tc-gluconate makes insoluble fraction only with H2S but not with other reactive kind sulfur compounds such as glutathione, cysteine, sulfite, sulfate, thiosulfate, or NO (ref [19], Figure 2).
2) It is recommended to provide biodistribution data for whole body. Nonspecific accumulation in normal organs and tissues, if any, might raise concerns about specific imaging.
ANSWER: We are sorry that we didn’t perform biodistribution study in this experiment. Nonspecific accumulation in normal organs and tissues can be found in our previous report (ref [19], Figure 4)
3) Figure 4B. It is difficult to see fluorescent intensity from images. It is recommended to provide quantified values of fluorescent intensity.
ANSWER: We are sorry for missing quantitative data. We had a problem with the computer linked to confocal microscope which contains raw data. It will take several weeks more to fix it. Anyway, the images were taken with the same condition at the same day. So, the intensities of the images might be comparable. Figure 4A with wider view also shows that the uptake in ischemic cells is higher than in normoxic.
4) Please provide ITLC data of 99mTc-gluconate and 99mTc-DTPA without NaHS.
ANSWER: We added the data in Figure 1.
5) Please discuss comparison of H2S with other hypoxia markers.
ANSWER: We think that hypoxia marker and H2S imaging might be overlapping significantly, because H2S is generated in many hypoxic tissues. However, we cannot say that they can be used vice and versa yet, because more experimental results are required.
6) Values described in Abstract are missing in the main manuscript. Eg, 'up to 40%' and 'SUVmean'. Please also include these values in the main manuscript.
ANSWER: Actually, “up to 40%” in Abstract was mistake and we removed it. “SUVmean values are added in the main text page 6 line 214-215.
7) Line 82. ‘99mTc’ is not expressed correctly. Also in other sections such as Line 284. Please check.
ANSWER: We corrected the typographic errors.
8) Symbol ‘micro’ should be expressed correctly, not ‘u’.
ANSWER: We corrected the typographic errors.
9) Figure 1. Please include unit of vertical axis.
ANSWER: We added “CPM” in the Figure 1.
10) Line 109. Figure 1, not Figure 11.
ANSWER: We corrected the typographic error.
11) Why did the author use CT26 cell line? Are there any specific reasons? Please comment.
ANSWER: We have been using CT26 cell line for in vitro and in vivo hypoxia experiments for many years. CT26 is easy to handle for us, and showed good reliable experimental results.
12) Cell uptake should be corrected (divided) by total radioactivity applied to cells. The values are generally expressed as ‘% initial activity/mg protein’.
ANSWER: We are sorry that we failed to calculate the % initial activity/mg because we don’t have total radioactivity information by mistake. Anyway, the CPM/mg data show relative uptakes, which might be enough for comparison of the cell uptakes.
13) Line 119. Figure 2, not Figure 7.
ANSWER: We corrected the typographic error.
14) Lines 125-127 seem to belong to a caption of Figure 2. Please move the sentences to the correct position.
ANSWER: We corrected the typographic error.
15) Figure 3A. Some words are hidden by images. Please check.
ANSWER: We replaced the image Figure 3A.
16) Figure 3A and 5A. Please include an explanation what a yellow triangle indicates.
ANSWER: “Yellow triangle represents hypoxic region” is added to Figure 3A. “(yellow triangle)” is added after “…… tourniquet-induced ischemia modeling after 3 h or reperfusion”.
17) Names of devices do not have to be provided in the result section. Include them only in the method section.
ANSWER: The names of devices were removed or moved to Materials and Methods section.
18) Line 205. Figure 5B, not Figure 10B.
ANSWER: We corrected the typographic error.
19) Figure 5A and B. Clear which leg is normal or ischemic.
ANSWER: Yellow triangles represent ischemic legs as corrected by above 16) comment.
20) Lines 211-212. Did the author intravenously inject the radiotracer directly into legs? Was that not via tail vein?
ANSWER: We corrected it by changing “via tail vein”.
21) Line 277. 99Mo/99mTc-generator, not 99M/99mTc-generator.
ANSWER: We corrected the typographic error.
22) In the discussion section, some repetitions of the introduction section are found. Eg, about disadvantages of fluorescent probes and advantages of 99mTc. Please remove.
ANSWER: We removed the sentence “However, there are disadvantages to using fluorescence …… to overcome these problems”.
23) Line 247. Fig 4 seems to be incorrect.
ANSWER: We corrected it into Figure 1.
24) Lines 330-331. Include a manufacture name of a BCA assay (kit).
ANSWER: We added that BCA protein assay kit was purchased from Pierce Co.” in Materials and Methods section. We greatly appreciate the reviewer for considerate and detailed comments.
Round 2
Reviewer 3 Report
All comments have been addressed.
However, in Figure 3 it seems that panel A overlaps with panel B. Please check.
I recommend this manuscript for publication after this is revised.